# The Medical Referral Process and Motor-Vehicle Crash Risk for Drivers with Dementia

**DOI:** 10.3390/geriatrics5040091

**Published:** 2020-11-13

**Authors:** Jonathan Davis, Cara Hamann, Brandon Butcher, Corinne Peek-Asa

**Affiliations:** 1University of Iowa Injury Prevention Research Center, 2190 Westlawn, Iowa City, IA 52242, USA; cara-hamann@uiowa.edu (C.H.); brandon-butcher@uiowa.edu (B.B.); corinne-peek-asa@uiowa.edu (C.P.-A.); 2Department of Epidemiology, College of Public Health, University of Iowa, 145 North Riverside Drive, Iowa City, IA 52242, USA; 3Department of Biostatistics, College of Public Health, University of Iowa, 145 North Riverside Drive, Iowa City, IA 52242, USA; 4Department of Occupational and Environmental Health, College of Public Health, University of Iowa, 145 North Riverside Drive, Iowa City, IA 52242, USA

**Keywords:** driver licensing, fitness to drive, dementia, Alzheimer’s disease, motor vehicle collision, driver evaluation

## Abstract

Cognitive and physical impairment can occur with dementia and reduce driving ability. In the United States, individual states have procedures to refer and evaluate drivers who may no longer be fit to drive. The license review process is not well understood for drivers with dementia. This study uses comprehensive data from the Iowa Department of Transportation to compare the referral process for drivers with and without dementia from January 2014 through November 2019. The likelihood of failing an evaluation test was compared between drivers with and without dementia using logistic regression. The risk of motor-vehicle crash after referral for review of driving ability was compared using a Cox proportional hazard model. Analysis controlled for the age and sex of the referred driver. Drivers with dementia performed worse on all tests evaluated except the visual screening test. After the referral process, the risk of crash was similar between those with and without dementia. Drivers with dementia were denied their license more frequently than referred drivers without dementia. However, drivers with dementia who successfully kept their license as a result of the license review process were not at an increased risk of crash compared to other referred drivers.

## 1. Introduction

In the United States, around five million adults aged 65 and older are living with dementia. The number of people with dementia is expected to double by 2060 [1]. Dementia impairs cognitive and physical ability, and these deficits can increase the risk of crashing while driving a motor vehicle. However, dementia has great variability in onset and progression of symptoms. Currently, there is little guidance on exactly at what point in the disease progression that the driver’s license should be restricted or denied. Cognitive impairment has been linked to poor performance on screening tests used to identify decreased driving ability and on driving examinations [2,3,4,5,6,7]. Despite this decreased performance, studies of medical records linked to licensing records fail to attribute a higher crash risk to drivers diagnosed with dementia [6,8,9].

A limitation of previous linkage studies is the inability to account for drivers who have had their licenses restricted or removed. Every state has procedures for referring a driver for a license review, but information about the effectiveness of these review processes is limited. The goal of the referral process is to accurately identify with high sensitivity and specificity as to which drivers need to limit or end driving to remove high risk drivers, but to allow safe drivers with mobility as long as possible. The process involves several steps beginning with the referral of drivers deemed to be at high risk and in need of a driving assessment. Referrals can be through law enforcement, licensing officials, healthcare providers, family members, the individual themselves, or based on an event such as a crash. Once a referral is made, the driver is assessed through various means that include cognitive tests, physical tests (e.g., vision, reaction time), driving tests, and medical evaluations. Based on testing results, the licensing official decides for no change, restriction, or denial of the license.

Iowa systematically tracks all drivers who are referred for review of their license through a system called the Enhanced Medical Referral and Evaluation Management System (EMREMS) and links outcomes with crashes and traffic convictions. A driver’s source of referral, tests administered during the review, and ultimate license disposition are tracked within EMREMS. In this study, we compared the license referral and review process for drivers with dementia and drivers without dementia. We hypothesized that drivers with dementia would perform worse on the driving evaluations used during the license review process and would be more likely to be denied their license. Additionally, we followed drivers after the license disposition to determine if the risk of being in a motor vehicle crash or receiving a traffic citation was different for referred drivers with and without dementia.

## 2. Materials and Methods

### 2.1. Study Design and Population

A retrospective cohort design was used to analyze the administration of driving evaluation tests, test outcomes, and the risk of driving outcomes among drivers with and without dementia who are referred for a medical review of their license. The driving outcomes included risk of crash and the risk of receiving a traffic conviction after being referred for review of a driver’s license. A medical review of the license is any request to the licensing authority to review someone’s ability to drive and maintain their license. Iowa requires in person renewal of a license for drivers over the age of 70 and every other issuance of the license for drivers under the age of 70. Drivers under the age of 70 were excluded from this study because of the differences in the renewal process. Iowa drivers over the age of 70 who had completed a medical review of their license between 1 January 2014 and 12 November 2019 were included in this study. Drivers were identified through Iowa’s Enhanced Medical Referral and Evaluation Management System (EMREMS). EMREMS is a comprehensive database that tracks all information of a person’s initial referral, the review process, and the ultimate disposition of their license. EMREMS was developed by the Iowa Department of Transportation (IDOT) and TransAmerica, LLC and fully implemented in the fall of 2015 with some records from 2014 entered retrospectively. IDOT has linked EMREMS data with licensure data concerning crashes and convictions. Any crash or conviction associated with a license is subsequently added to the EMREMS record, so crashes and convictions can be identified after the review process is complete. Follow-up for crash risk started at the date the driver was given their license disposition. Drivers were followed until 12 November 2019 or the date of their crash. Drivers were excluded if they died during the review process. Only drivers who completed the review process and received a disposition on their license were included in the analysis of administration and performance on screening tests. Drivers who voluntarily surrendered their license were excluded as they could not be given evaluation tests if no longer participating in the review process. The data used for analysis were de-identified by the IDOT. The procedures of this study were reviewed by the University of Iowa Institutional Review Board. The project was determined to be not human subjects on 21 June 2018 (Project #201805949).

### 2.2. Variables

Drivers with dementia were identified by the presence of the diagnosis of dementia or Alzheimer’s disease on a medical report completed as part of a request by the Iowa DOT or by a physician letter indicating the driver had dementia and should have their license reviewed or suspended. EMREMS captures the results of several tests used as part of the medical review of a license. These tests include the following: Driver Orientation Screen for Cognitive Impairment (DOSCI), Safe Driving BASICS (Brief Auto-Screening Instrument for Cognitive Status), on-road driving test, driving knowledge test, vision screening, and medical report of fitness-to-drive. The DOSCI and Safe Driving BASICS are cognitive screening tests. The DOSCI asks the individual about themselves (name, home address) and where they are currently (location, time). There are nine total questions asked in the DOSCI and a failure of three or more is considered inadequate by the Iowa DOT. The DOSCI has been used to identify individuals with possible cognitive impairment during roadside assessments [10]. Safe Driving BASICS is a computer-administered testing battery that combines aspects of trail-making, visual memory, and visual closure (ability to visualize missing information). Each component of the Safe Driving BASICS has a standard time in which that component should be completed or a set number of acceptable incorrect responses. A failure of any one of the tests that make up Safe Driving BASICS would be considered inadequate by the Iowa DOT. The individual tests that make up Safe Driving BASICS identify older drivers whose scores place them in a group at statistically higher risk of a crash [2,5,11,12,13]. The on-road driving test is the primary determinant of a person’s ability to continue driving. Failure of the on-road driving test occurs when a driver violates a law, causes a crash, takes a dangerous action, or accumulates points for driving errors. The same driving standards are used for each driver and are administered by a trained evaluator.

An individual can be referred for a medical review of their license by several different groups. We described all sources that accounted for more than 1% of referrals. These sources include the following: the driver themselves, law enforcement, physicians, IDOT licensing official during renewal exam, recall based on a previous encounter, and as a result of a crash. IDOT licensing officials can refer drivers for a review of their license as part of license renewal if the driver appears to have diminished driving abilities (Iowa Administrative Code 604.50(5)). The official observes the applicant for impairment and asks about changes in medical conditions that may affect the applicants driving ability. Periodic recall of a driver to have their license reviewed can be required by the Iowa DOT before a driver can renew their license. This occurs based on a previous license renewal experience. The other sources of referrals that did not make up 1% of the total referrals included court/hearing requests, public letter for review, family, or administrative request.

Outcome variables included crashes and traffic convictions. Crashes were any crash reported by law enforcement in the state of Iowa or crashes requiring an accident report. An accident report is required for crashes that result in death, personal injury, or total property damage of $1500 or more. Traffic convictions include any traffic enforcement conviction issued by an Iowa law enforcement officer for a moving violation and reported to the Department of Transportation. A moving violation occurs while a car is in motion and a traffic law is violated. Examples include speeding, disobeying traffic signs, or operating while intoxicated. All moving violations are identified in Iowa Administrative Code 761-615.1. Non-moving violations are not reported to the Department of Transportation and include violations related to parking, equipment standards, or registration. For this study, we included only moving violations.

### 2.3. Statistical Analysis

The proportion of drivers referred for medical review of their license was described for those with and without a diagnosis of dementia. Age and sex were also described across the diagnosis of dementia. For drivers that did not voluntarily withdraw from the license review process, the screening tests given as part of the review were compared across drivers with and without dementia. Multivariable logistic regression models were built to assess the odds of failure of the various tests across the two groups, controlling for age and sex. The covariates for age and sex were chosen for inclusion in the models a priori. Age was included in the model because risk of dementia increases with older age and older age can result in other impairments that reduce driving ability [14,15,16]. Sex was included based on the differences in self-regulation and driving cessation that occurs across male and female drivers. Since male and female drivers approach driving retirement differently, the population of referred drivers may differ in driving ability across sex [17].

The driver’s risk of crash post review was evaluated using Cox proportional hazard models controlling for five-year age group and sex. Licensing status will affect future driving exposure. If a driver is denied their license, they will likely drive less in the future. To control for the difference in driving exposure, the analysis was completed separately for driver’s denied their license and those who received their license. In addition to the crash outcome, hazard ratios were calculated for traffic convictions using the same methods. All analyses were completed using SAS 9.4 (SAS Institute, Cary, NC, USA). An alpha level of 0.05 was used for tests of significance.

## 3. Results

There were 21,599 drivers over the age of 70 included in EMREMS from 1 January 2014 and 12 November 2019 who had a medical review of their driver’s license and either completed the review process or voluntarily surrendered their license. Of the drivers, 574 (2.7%) had a diagnosis of dementia. Table 1 presents the demographics and referral sources for drivers with and without dementia. Drivers with dementia were more likely to be male (61.6% vs. 49.4%, *p* < 0.001) when compared to other referred drivers. Drivers with dementia were slightly younger with a median age of 82.0, while drivers without dementia had a median age of 82.9. The difference in age is driven by a higher proportion of drivers aged 90 and older amongst those without a diagnosis of dementia (13.2% vs. 7.1%, *p* < 0.001). The two groups of drivers differed greatly in terms of the sources of referral for medical review of their license (Table 1). Drivers with dementia were more likely to be referred by a physician (18.3 vs. 1.6, *p* < 0.001) or law enforcement (13.9 vs. 8.5, *p* < 0.001). Drivers without dementia were more likely to be referred for license review as a result of a crash review (19.2% vs. 9.4%, *p* < 0.001) compared to drivers with dementia. For both groups, referral was most frequently initiated by a DOT licensing official as part of the license renewal process.

Of the referred drivers, 16,879 (80.2%) drivers without dementia and 503 (87.6%) drivers with dementia completed the license review process and received a disposition for their license without voluntarily surrendering their license. The tests given during the license review and their results are described for both groups in Table 2. The differences across the diagnosis of dementia for sex and age remained in the drivers who completed the review process. The types of tests administered to drivers with and without dementia varied significantly. Except for requiring a medical report, drivers without dementia were more frequently given each of the other tests used to evaluate driving ability and knowledge. The greatest difference in the frequency of a test being given between the two groups was the on-road driving test, for which 50.1% of drivers without dementia completed an on-road driving test compared to 29.0% of drivers with dementia (*p* < 0.001). There was also a large difference in the administration of the DOSCI, for which drivers without dementia were more likely to be given this screening test (72.1% vs. 55.2%, *p* < 0.001). Less testing is partially explained by some drivers with dementia being initially identified and referred by a medical professional. Many of these drivers would not be screened with the DOSCI if they already had a diagnosis of dementia on their record. The ultimate license disposition of drivers with and without dementia is also presented in Table 2. A higher proportion of drivers with dementia were denied their license compared to drivers without dementia (69.2% vs. 10.3%, *p* < 0.001). Drivers, on average, received a disposition about six weeks after the review process began.

Logistic regression was used to compare the likelihood of failing the various driving-related tests among those with dementia compared to those without dementia (Table 3). Drivers with dementia were significantly more likely to fail all tests except the vision screening test. The greatest increase in the odds of failing a test occurred for the DOSCI (OR = 42.37, 95% CI: 32.46–55.30), where drivers with dementia were significantly more likely to fail.

The Cox proportional hazard models for the risks of being in a crash and receiving a traffic conviction after completion of license review are given in Table 4 separately. Two models were provided for each outcome, a model for drivers denied their license and those who were able to legally continue driving with or without a restriction added to their license. Of those who retained their driver’s license, drivers with dementia had a slightly higher risk of a crash following completion of their license review (HR = 1.25 95%, CI: 0.69–2.27), but this result was not statistically significant. There was no difference in the risk of receiving a traffic conviction post-license review between drivers with and without dementia. Similar results were found for drivers who were denied their license, but a smaller proportion of these drivers experienced a crash or traffic conviction.

## 4. Discussion

Among drivers over 70 undergoing a license review, those who had dementia were far more likely to have their license denied than those without dementia (69.2% compared to 10.3%). The likelihood of denial should alert family members, physicians, and the drivers themselves that a review of driving ability should coincide with diagnosis and progression of dementia. High denial rates for licenses is consistent with other research of drivers with dementia. In an evaluation of Oregon drivers referred by physicians, very few drivers were able to keep their license [18]. In our data, we were able to examine multiple types of referral sources in addition to referrals from physicians, and license denial for those with dementia was common regardless of referral source (data not shown).

Drivers with dementia were much less likely to be given an on-road driving test or knowledge test than drivers without dementia. Instead, drivers with dementia were more frequently required to provide a medical report or were referred by a physician through a medical report. The decision to deny the license for many drivers with dementia was based on the medical report or request of their physician. In most cases, these drivers would require a subsequent medical report indicating they are fit to drive before being granted their license or further testing of their driving ability.

When drivers with dementia were evaluated with the various screening tests, they were significantly more likely to fail every test except for the vision screen. Previous studies suggest that the use of cognitive screening tests can predict a person’s ability to pass an on-road driving test [7]. In our study, drivers with dementia were less likely to pass IDOT’s screening battery (the Safe Driving BASICS) as well as the individual DOSCI test and the on-road driving test. The DOSCI was developed to identify drivers with cognitive impairment [10]. As expected, drivers with dementia had a much higher likelihood of failing the DOSCI. A driver without dementia passing the DOSCI does not imply that the driver is fit-to-drive. Tests more specifically evaluating driving ability are needed to make licensing decisions for drivers who pass the DOSCI such as an on-road driving test.

The increased likelihood of failing the various screening and driving tests supports that the referral of drivers with dementia for license review is necessary. As license denial for many of these drivers is based on the recommendation of a driver’s physician, those with dementia who go on to complete the Iowa DOT tests may be those drivers with less advanced dementia. For these drivers, a physician may be requesting further evaluation of their driving ability rather than recommending outright denial of the license. The drivers with dementia who are evaluated are expected to be more capable of passing the tests during review of their license than those who were denied their license based on the physician’s recommendation. This would bias performance on the evaluation tests toward finding no difference between drivers with and without dementia. However, we found an increased likelihood of failure of all tests, except the vision screening test. If all referred drivers with dementia were given these tests, we would expect the odds of failure to be greater than that found in our study.

Determining a specific guide to making a licensing or referral decision was beyond the scope of this study. However, we observed that drivers with dementia who went on to maintain their license had passed the on-road driving and knowledge tests. When we evaluated the crash risk of drivers with dementia who kept their license compared to drivers without dementia, we found an elevated but not statistically significant crash risk. Future studies with an increased follow-up period of drivers referred for medical review are needed to provide evidence of the effectiveness of the driver’s license review process for individuals with dementia.

We found referred drivers with dementia were more likely to be male. This may be a result of male drivers less frequently ceasing from driving as dementia progresses than female drivers. A meta-analysis of studies evaluating driver cessation across men and women found women were more likely to self-regulate or retire from driving [17]. Alternatively, the higher number of male drivers could be the result of the referral sources more frequently referring male drivers, but supporting data for this hypothesis does not exist.

When comparing how drivers were referred for review, physician referral was much more frequent for drivers with dementia compared to those without dementia. In Iowa, physician referral is voluntary, and states vary in having voluntary and mandatory reporting. One previous study found no differences between mandatory and voluntary reporting laws in reducing dementia prevalence among motor vehicle crash-related hospitalizations, but did find the in-person renewal laws reduced dementia prevalence among those hospitalized [19]. We found that for those referred, drivers with dementia were much more likely to have their license denied, so we would expect states with mandatory physician reporting would have fewer crash injuries for drivers with dementia. However, the number of individuals referred by physicians is relatively small. Most drivers with dementia were referred by non-physician sources with licensing officials being the most common source for review.

Studies linking licensure, medical record, and crash data have consistently found either no increased risk of a crash [6,9] or a decreased risk of a crash for drivers with dementia [8]. The previous studies rely on license status as a means of identifying drivers. In addition to any precautions taken by the driver, removal of driving privileges of those at greatest risk may contribute to this decreased crash risk found in linkage studies. In our study, we demonstrated that drivers with dementia referred for medical review of their license were likely to have their license denied. Those with dementia who were able to continue driving performed well on the driving evaluations. Since many drivers can continue driving safely during the early onset of dementia [3,20], it is not surprising that studies solely using licenses as a surrogate for driving exposure failed to find a difference in crash risk for drivers with and without dementia. Those still driving are the least likely to have their driving ability impacted by dementia. In our study, we did not have driving exposure post referral, but we found no significant difference in crash or traffic convictions after the referral process was completed between drivers with and without dementia. From the available data, we could not identify if the drivers with dementia who kept their license were as fit-to-drive as drivers without dementia or if they were simply driving less. Further study accounting for the driving habits after review of a license would provide essential information about safe driving during the early stages of dementia.

The data from EMREMS used for this study were an existing dataset that tracks the review process for referred drivers. There are some limitations to using these data. We sought to describe the difference in the review process for drivers with dementia compared to drivers without dementia. Not all drivers required a medical report or were referred by physicians, so some drivers with dementia will have been categorized as dementia-free. To minimize any bias this would introduce, we limited our analysis of performance on screening tests and driving evaluation tests to drivers who completed the review process. This prevents the misclassification of drivers who surrender their license and are unable to be evaluated for dementia. Additionally, not all drivers with dementia are referred for license review. The drivers referred are likely to be those with the more advanced progression of dementia, resulting in a higher likelihood of referral. This is evident by the poor performance on driving evaluations exhibited by those with dementia in our study. Additional studies evaluating the progression of dementia and performance on driving evaluations are needed to further inform the fitness-to-drive evaluation.

While limitations exist for the use of secondary data, it allows for a deeper understanding of the current referral process. Additionally, we were able to include drivers referred by any source. The majority of previous evaluations of license review data rely solely on referrals from law enforcement [21,22,23]. The use of a comprehensive tracking system is especially important for the evaluation of drivers with dementia since very few are referred specifically by law enforcement.

## 5. Conclusions

Drivers with dementia referred for medical review are rarely allowed to continue driving and performed worse than other referred drivers on driving evaluation tests. Physicians are an important source for the referral of drivers with dementia, but most drivers with dementia were referred by other sources. Most referrals came from IDOT licensing officials during license renewal. Our results support that in-person renewal plays a pivotal role in this identification process since most referrals, dementia or no dementia, were the result of in person license renewal. Guidelines on who should be referred for license review are available [24], but concede that the circumstances of the individual driver are required in making a referral. We found that once referred, drivers with dementia performed much worse on driving evaluations. Based on the high rate of denial, it appears that sources of referral are being conservative with which drivers with dementia are referred and only those with the most impaired driving are being evaluated. Self-restriction of driving is widespread among older adults with diminished abilities and helpful in preventing motor vehicle crashes. Self-restriction, however, is not a substitution for a formal review of driving ability. Family members, physicians, and the driver themselves can utilize license review for guidance on appropriate driving habits after diagnosis with dementia.

## Figures and Tables

**Table 1 geriatrics-05-00091-t001:** Demographics and referral source of drivers.

		No Dementia		Dementia		
Variable	Level	N = 21,025	%	N = 574	%	*p*-Value ^a^
Sex ^b^	Female	10,630	50.6	220	38.4	
	Male	10,391	49.4	353	61.6	<0.001
Age Group	70–74	2872	13.7	63	11.0	
	75–79	4172	19.8	126	22.0	
	80–84	6282	29.9	206	35.9	
	85–89	4932	23.5	138	24.0	
	90+	2767	13.2	41	7.1	<0.001
Line Exam		10,665	50.7	254	44.3	0.002
Recall/Periodic Testing		2666	12.7	22	3.8	<0.001
Crash Review		4038	19.2	54	9.4	<0.001
Law Enforcement		1778	8.5	80	13.9	<0.001
Driver Self		1209	5.8	22	3.8	0.051
Physician		336	1.6	105	18.3	<0.001
Other		333	1.6	37	6.4	

^a^ The *p*-value is calculated by chi-square test. ^b^ Values do not add to column total due to missing values.

**Table 2 geriatrics-05-00091-t002:** Differences in administration of the test to drivers with and without dementia.

		No Dementia		Dementia		
Variable	Level	N = 16,879 ^a^	%	N = 503 ^a^	%	*p*-Value ^b^
Sex	Female	8408	49.8	193	38.4	
	Male	8471	50.2	310	61.6	<0.001
Age Group	70–74	2409	14.3	59	11.7	
	75–79	3465	20.5	107	21.2	
	80–84	5149	30.5	181	35.9	
	85–89	3828	22.7	119	23.6	
	90+	2028	12.0	38	7.5	0.004
On-road Driving Test	Not Tested	8428	49.9	358	71.0	
	Tested	8451	50.1	146	29.0	<0.001
Knowledge Test	Not Tested	12,794	75.8	417	82.7	
	Tested	4085	24.2	87	17.3	<0.001
Safe Driving BASICS	Not Tested	15,249	90.3	464	92.1	
	Tested	1630	9.7	40	7.9	0.197
DOSCI	Not Tested	4705	27.9	226	44.8	
	Tested	12,174	72.1	278	55.2	<0.001
Medical Report	Not Tested	13,709	81.2	95	18.8	
	Tested	3170	18.8	409	81.2	<0.001
Vision Screening	Not Tested	10,189	60.4	324	64.3	
	Tested	6690	39.6	180	35.7	0.076
Disposition	No Change	7777	46.1	28	5.6	
	Restriction Change	7364	43.6	127	25.2	
	Denial of License	1738	10.3	349	69.2	<0.001

Abbreviations: BASICS, Brief Auto-Screening Instrument for Cognitive Status; DOSCI, Driver Orientation Screen for Cognitive Impairment. ^a^ Excludes drivers who surrendered their license. ^b^ The *p*-value is calculated by chi-square test.

**Table 3 geriatrics-05-00091-t003:** Odds ratio of failing tests for those with dementia compared to those without dementia.

Test	Odds Ratio (95% CI) ^a^
On-Road Driving Test	3.30 (2.20–4.94)
Knowledge Test	7.53 (4.49–12.64)
Safe Driving BASICS	6.54 (2.83–15.07)
DOSCI	42.37 (32.46–55.30)
Medical Report	9.80 (7.82–12.27)
Vision Screening	0.38 (0.14–1.03)

Abbreviations: BASICS, Brief Auto-Screening Instrument for Cognitive Status; DOSCI, Driver Orientation Screen for Cognitive Impairment. ^a^ Controlling for five-year age group and sex.

**Table 4 geriatrics-05-00091-t004:** Hazard ratios for crash and conviction post license review.

Outcome	Diagnosis	Hazard Ratio (95% CI) ^c^
Crash–Denied ^a^	Dementia	1.26 (0.63–2.50)
	Not Diagnosed	-
Crash–Licensed ^b^	Dementia	1.25 (0.69–2.27)
	Not Diagnosed	-
Traffic Conviction–Denied ^a^	Dementia	1.00 (0.56–1.76)
	Not Diagnosed	-
Traffic Conviction–Licensed ^b^	Dementia	1.06 (0.53–2.14)
	Not Diagnosed	-

^a^ Denied indicates analysis of drivers denied their license comparing drivers with and without dementia. ^b^ Licensed includes drivers who were given their license with no change or a restriction but were able to continue driving. ^c^ Controlling for five-year age group and sex.

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
