# Peer review of "The Medical Referral Process and Motor-Vehicle Crash Risk for Drivers with Dementia"

_geriatrics, 2020, doi:10.3390/geriatrics5040091_

Round 1
Reviewer 1 Report
This article addresses an important issue on the medical referral process for older drivers with dementia. Articles of this magnitude (with a large sample) that allow for compelling demonstrations are really useful. The database from which the authors conducted their analysis is really interesting, rich and the results are convincing. It allows to put into perspective the decision after the referral process and the risk of crash of the drivers.
Points to be reviewed :
In my opinion, the article deals more deeply with the issue of medical referral than with the risk of drivers with dementia. I would therefore reverse these two elements in the title: "Medical Referral Process and Motor-Vehicle Crash for ..."
A small criticism of the hypothesis that I find really obvious. Perhaps rephrase it in a more challenged way ? the second prediction about the drivers following is more original.
Line 101. "The individual tests 101 that make up Safe Driving BASICS identify older drivers whose scores place them in a group at 102 statistically higher risk of a crash". It would be useful to clarify how decisions are made when participants get values very close to the cut-off points. Is the decision discussed, other tests proposed? The authors could raise this point in discussion as a limitation of the approach (related to the sensitivity/specificity issue).
Line 196. "The Cox proportional hazard models for the risks of being in a crash and receiving a traffic conviction are given in Table 4 separately". It is not clear to me whether this risk of being in a crash concerns the period before or after the licence review. The information is given 5 lines below but perhaps it should be indicated earlier?
Two analyses could have been conducted if the data were available: one before and one after the licence review or medical referral.
Line 202. "...drivers who were denied their license, but a smaller proportion of these drivers experienced a crash or traffic conviction". My problem of comprehension is the same as in the previous point. When did these drivers stop and when did they have these accidents (before or after the medical referral?) it's a bit vague...
Table 3. The on-road test is often perceived as the gold standard of driving assessment. In this study, even if the Odds ratio is high, there are many other tests that are even more important to take into account in the evaluation. The DOSCI in particular deserves to be highlighted more in the conclusion?
Reviewer 2 Report
This manuscript uses administrative data from the Iowa Department of Transportation to describe the process for older adult drivers who have been referred for medical review prior to license renewal, the results of the review and subsequent rate of motor vehicle crashes and traffic citations.
Three hypotheses are stated at the end of the Introduction: (1) drivers with dementia perform worse on evaluations; (2) drivers with dementia are more likely to be denied their license; and (3) risk of motor vehicle crash or traffic citation differs for referred drivers with versus without dementia. The second hypothesis is relatively simple and straightforward to test (and is confirmed). The adequacy of the secondary data for hypotheses one and three is more problematic.
Major concerns regarding the first hypothesis include the lack of detail regarding the assessments, e.g.
- What is the definition of failure for these tests?
- Has the same definition been used by all examiners?
- Understanding that this is not a formal study of the evaluation process, does knowledge of a driver's dementia status influence the examiner's judgment in the evaluation (i.e. bias via unblinded assessment)?
- Perhaps most importantly, since each type of evaluation is done for only a subset of drivers in each dementia category, how might that selection process affect the results?
The authors acknowledge the limitations of the data for the third hypothesis - the absence of information on actual driving exposure. This is a significant weakness.
I also have some specific questions and comments. The first sentence of the Variables section indicates that dementia is defined by a medical report provided by a physician. If so, why is "medical report" listed as a test in Table 2 and why is it only present for 81.2% of drivers with dementia (instead of 100%)?
In Table 1, the sum of all sources of referral does not equal the total number. For example, for drivers with dementia there's a referral source for 515 of the 574 drivers. Are some drivers reviewed without having been referred?
In Table 4, it's surprising that the hazard ratios for crashes are virtually identical (1.26 and 1.25) and that the width of the confidence intervals are similar despite a much larger number of licensed and non-diagnosed drivers. Those values should be confirmed.
The end of the first paragraph of the Discussion states that license denial for drivers with dementia was similar for all sources of referral. This was not shown in the Results, so a "data not shown" comment could be included with this statement.
One possible misspelling - on page 3, line 130, should "cession" be "cessation"/
Round 2
Reviewer 2 Report
The authors have been very thoughtful and responsive to my comments on the initial version of this manuscript.